# Durum Wheat–Potato Crop Rotation, Soil Tillage, and Fertilization Source Affect Soil $CO_2$ Emission and C Storage in the Mediterranean Environment

Roberto Mancinelli [1], Sara Marinari [2], Mariam Atait [1], Verdiana Petroselli [1], Gabriele Chilosi [2], Merima Jasarevic [2], Alessia Catalani [2], Zainul Abideen [3], Morad Mirzaei [4], Mohamed Allam [1,*] and Emanuele Radicetti [5]

1    Department of Agriculture and Forest Sciences (DAFNE), University of Tuscia, I-01100 Viterbo, Italy
2    Department for Innovation in Biological, Agro-Food and Forest Systems (DIBAF), University of Tuscia, I-01100 Viterbo, Italy
3    Dr. Muhammad Ajmal Khan Institute of Sustainable Halophyte Utilization, University of Karachi, Karachi 75270, Pakistan
4    Department of Soil Science and Engineering, Faculty of Agricultural Engineering and Technology, University of Tehran, Karaj 31587-77871, Iran
5    Department of Chemical, Pharmaceutical and Agricultural Sciences (DOCPAS), University of Ferrara, I-44121 Ferrara, Italy
*    Correspondence: mohamed.allam@studenti.unitus.it

**Abstract:** At present, the role of agricultural practices on the dynamic of GHGs is being investigated worldwide. In this study, the effects of soil tillage practices (conventional vs. conservation techniques) and fertilization sources (inorganic vs. organic) on soil $CO_2$ emissions in durum wheat (*Triticum durum* Desf.)–potato (*Solanum tuberosum* L.) rotation in the Mediterranean area were evaluated. This study aimed to understand the changes in the soil carbon content and the soil $CO_2$ emissions under different soil tillage practices (moldboard plow (P), subsoiler (R), and spading machine (S)) and fertilization sources (mineral (M) and organic (O) with municipal organic waste). Soil $CO_2$ flux, soil water content, and soil temperature data were collected for both crops using a portable closed-chamber infrared gas dynamics system. Significant relationships were detected between soil $CO_2$ emissions and soil temperate and soil water content. However, these relationships were found only for durum wheat crops. Our findings indicate that including sustainable agricultural practices in wheat–potato rotation system could act as an appropriate alternative option to increase soil organic carbon, mitigate $CO_2$ emissions, and reduce the dependence on chemical inputs and energy.

**Keywords:** $CO_2$ emission; global warming; sustainable agriculture; wheat-potato crop rotation

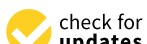



## 1. Introduction

Rapid population growth associated with several challenges, such as lack of food security and energy, increased emissions of greenhouse gases (GHGs), climate change, and soil degradation, have adversely impacted lives on earth [1–5]. $CO_2$ is a major GHG that significantly affects global warming and climate change [6,7]. Anthropogenic activities significantly affect the emissions of GHGs and global warming [8,9]. Within the framework of the European Union, the aim is to become climate neutral by 2050. This plan, therefore, foresees the need to mitigate emissions by more than double the average reduction achieved each year between 1990 and 2020 [10]. The burning of fossil fuels, deforestation, intensive tillage operations, excessive use of nitrogen fertilizer, and burning of postharvest crop residues are among the main anthropogenic activities that strongly impact the emissions of $CO_2$ [11,12].

Around 10% of the total GHG emissions originate from the agriculture sector, with the largest share being attributed to emissions from cultivated land (5.27%), followed by

3.21% due to the intestinal fermentation of livestock, and others associated with the use of livestock manure (1.58%) [8,13]. The main inputs of carbon (C) into soil are through the decomposition of dead biomass, organic residues, and animal manure, while C losses mainly occur via roots and microorganisms' respiration [14]. Their relative emission flows depend on soil water content, soil temperature, nutrient availability, pH value, and land cover and management [15,16]. Agricultural activities play a crucial role in balancing the emissions of GHGs, especially $CO_2$ [6,17], Therefore, the development and adaptation of sustainable management practices in the agro-ecosystem is required to mitigate the emissions of GHGs and meet environmental concerns.

Soil tillage and fertilization practices play an important role in the dynamic of carbon and nitrogen and GHG emissions [18,19]. They represent the main agricultural practices implemented by farmers to secure high crop productivity. However, the conventional tillage system and the application of synthetic fertilizers can directly and indirectly impact environmental pollution [20,21]. Conservation tillage practices represent a fundamental pillar of sustainable farming systems [22], as plowing is the most energy-demanding process in the production of arable crops [23]. Conservation tillage techniques, such as reduced or minimum tillage (RT) and no-tillage systems, are gradually becoming reliable for farmers. In comparison with plowing, reduced tillage techniques decrease macro-aggregate destruction, and hence reduce the exposure of soil organic matter to mineralization [24]. Moreover, they improve soil health [25], enhance soil's physical properties [26], increase soil microbial biomass [27,28], positively affect soil aggregation [29], and significantly decrease greenhouse gas emissions [30,31]. Furthermore, under organic fertilization management, enhanced soil nutrient availability [32] and improved soil quality [33] have been reported. Wu et al. [34] suggested that using organic nutrient sources can be an effective practice to restore microbial biomass loss due to the intensive application of chemical fertilizers. Importantly, they play a positive role in climate change mitigation by soil C sequestration [32]. Using sustainable agricultural practices, such as conservative tillage techniques and the application of organic fertilizers, could be a solution to mitigate GHGs and improve crop performance. This study hypothesizes that reduced tillage practices and organic fertilization could represent a suitable and efficient strategy, in terms of carbon sequestration and reduction in greenhouse gas emissions, in a potato–durum wheat crop rotation system. In this research, we aimed to study the effects of three different tillage systems (plowing, subsoiling, and using a spading machine) and two types of fertilizer sources (inorganic vs. organic), on soil $CO_2$ emissions, soil organic C content, soil temperature and volumetric water content. In addition, the relationships between these soil parameters in a two-year durum wheat–potato rotation system typical of the Mediterranean area will be evaluated.

## 2. Materials and Methods

### 2.1. Study Site

The research was conducted at the "Nello Lupori" experimental farm of the University of Tuscia, central Italy (42°25′32″ N, 12°04′52″ E, at 310 m a.s.l.) in two growing seasons, 2021 and 2022 (Figure 1). The site is characterized by a typical Mediterranean climate with warm humid winters and hot dry summers. The average temperature is 14.5 °C, with minimum temperatures just below 0 °C in February and maximum temperatures above 35 °C in July. The average annual rainfall is 752 mm (last 30 years), mainly concentrated from September to May (569 mm). The soil of the experimental area is of volcanic origin and classified as *Typic Xerofluvent* [35]. The particle size distribution of the soil in the surface horizon (depth 0–30 cm) is defined by 76.3% sand, 13.3% silt, and 10.4% clay; therefore, it is sandy soil-clay. The soil had 0.15% total nitrogen (Kjeldhal method), 0.97% total organic content of C (Lotti method), a pH of 6.9 (water, 1:2.5), EC 2.5 dS m$^{-1}$, 33 mg kg$^{-1}$ available phosphorous (Olsen extraction method) and 575 mg kg$^{-1}$ exchangeable potassium (K+, extracted using 1.0 N neutral).

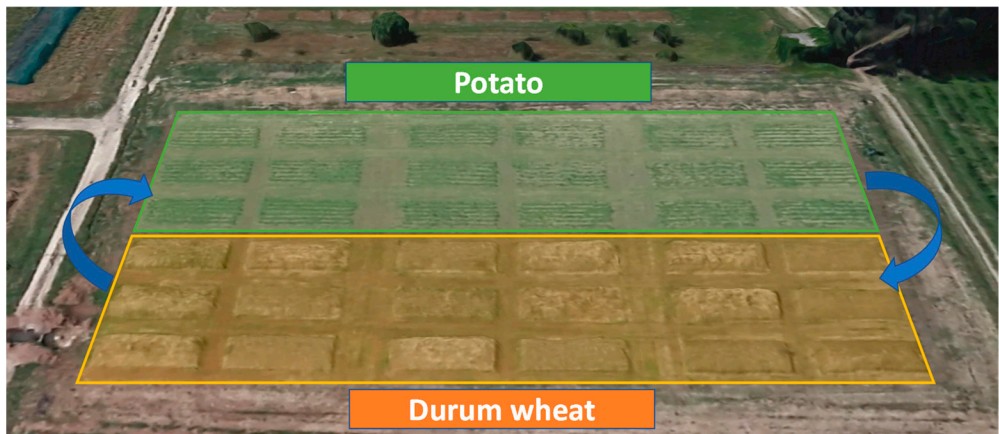

**Figure 1.** Experimental field plan of durum wheat–potato rotation.

## 2.2. Field Setup

The experiment was conducted on a biennial rotation, typical of the study area, between durum wheat and potato, to compare different tillage practices and fertilization events. The treatments consisted of: (i) three soil tillage systems (conventional with moldboard plow (P), reduced tillage with a subsoiler (R), and reduced tillage with a spading machine (S)) and (ii) two fertilization sources (mineral fertilizer (M) and organic (O) with municipal organic waste). The two main crops in rotation were simultaneously cropped every year. For each crop, the treatments were replicated three times according to a randomized complete block design. The experimental field included 36 plots (2 crops × 2 fertilization sources × 2 tillage managements × 3 blocks). Each plot had a size of 60 m$^2$ (6 × 10 m) and 5-m-wide alleys separated all plots of the experimental field in carrying out all farming operations.

The conventional tillage practice (P) consisted of moldboard plowing at 30 cm soil depth, while subsoil tillage (R) and a spading machine (S) were employed as minimum tillage practices at 20 cm depth. The durum wheat, variety Antalis, was sown in late November to early December in both growing seasons by means of a plot seeder at a sowing density of 400 grains per m$^2$. A potato crop, *Levante* variety, was sown manually with a density of 7.4 plants per m$^2$ on 8 April 2021 and 11 April 2022, respectively. The nitrogen requirement of wheat, in the mineral application, was divided into two applications, 50% in the tillering and 50% in stem elongation, for a total of 100 kg ha$^{-1}$ of N. The demand, in organic management, was met by 25 t ha$^{-1}$ of compost from municipal solid waste (ACEA Ambiente, Orvieto Italia). The mineral fertilization of the potato has been divided into two applications, one for sowing with 100 kg ha$^{-1}$ of P$_2$O$_5$, 50 kg ha$^{-1}$ of K$_2$O and 100 kg ha$^{-1}$ of urea, and a second during the tamping with 70 kg ha$^{-1}$ of ammonium nitrate. For organic fertilization, however, only one application was carried out before sowing with 18 t ha$^{-1}$ of compost from municipal solid waste (ACEA Ambiente, Orvieto Italia). The field setup and crop management are reported in detail by Mancinelli et al. [36]. The potato field was irrigated based on local weather conditions and irrigation was stopped two weeks prior to harvesting. The wheat was threshed on 14 July 2021, and 21 June 2022, and the potato was harvested on 9 August and 26 July for the first and second crop cycles, respectively.

## 2.3. Data Collection

Meteorological data for the 2021 and 2022 growing seasons were collected from a meteorological station 500 m from the experimental site. Soil temperature, soil water content, and soil CO$_2$ flux data were collected for entire crop cycles using a portable closed-chamber infrared gas dynamics analysis system following the methodology of Mancinelli et al. [37]. Measurements were collected every three weeks (±2 days) in each durum wheat growing season, while measurements were collected every two weeks (±2 days) in

each potato growing season. The measurements were always performed between 9 and 10 AM, and after 2 days in the case of rainfall events. A non-stationary flow-through chamber (SRC-1, PP Systems, Stotfold, UK) with a volume of 1334 $cm^3$ and a coverage area of 78.5 $cm^2$ was used to perform soil $CO_2$ emission measurements [37]. The soil $CO_2$ readings were taken by placing the instrument's reading bell in a permanent PVC cylinder positioned in a central area of each plot and kept free from any weeds. The amount of soil $CO_2$ emissions was estimated by fitting a quadratic equation to the ratio of the increase in $CO_2$ concentration to the elapsed time. The values obtained were considered the net carbon mineralization, since the autotrophic component was included. The estimate was determined by considering two consecutive measurements in linear interpolation and their integration over time (trapezium rule) as previously adopted by Mancinelli et al. [38]. The soil temperature was measured with an STP1 probe connected to the EGM-4 instrument at a depth of 5 cm and at the same time the volumetric water content of the soil was measured at a depth of 0–30 cm with a TDR 300 instrument (Spectrum Technologies, Inc. Plainfield, IL, USA).

For the analysis of total organic carbon (TOC)and total nitrogen (TN) in the soil, five samples were taken for each plot (0–20 cm soil depth), then the samples were air dried and passed through a 2-mm sieve and stored at a temperature of 4 °C. The TOC and TN of the soil samples were determined using an elemental analyzer (Thermo Soil NC-Flash EX1112). The C content in the epigeal samples of wheat and potato crops was also determined using the same elemental analyzer. The plant samples were taken in each plot, in the ripening state, on a double row of 1 $m^2$ for the wheat, and on a single row of 2 $m^2$ for the potato. Subsequently, they were dried in an oven at 70 °C until completely dry, and powdered. Lastly, the data obtained (C input) were related to the relative $CO_2$ emissions from the soil (C output) to determine the C input/C output ratio [39].

### 2.4. Statistical Analysis

All the data were analyzed using JMP 4.0 software. Two-way factorial analysis of variance (ANOVA) was applied to evaluate the effects of soil tillage practices and fertilizer source ad their interactions on $CO_2$ emissions and other soil parameters, where years were considered as a repeated measurements across time [40]. Fisher's test (LSD) was performed to examine the differences between mean values. Differences at $p < 0.05$ were considered statistically significant. Finally, a second-order polynomial regression was used to relate data on soil $CO_2$ emissions under different treatments (soil tillage and fertilization managements) with soil temperature and soil water content (Tables S1 and S2).

### 3. Results

*3.1. Weather Conditions during Durum Wheat–Potato Crop Rotation*

The 2020/2021 and 2021/2022 growing seasons differed greatly in terms of rainfall and temperature (Figure 2). The years 2020/2021 were wetter compared with 2021/2022 (770 vs. 177 m, respectively) except the period from mid-March to the beginning of April. Conversely, 2022 was dry or semi-arid during the whole season, except in April, when wet conditions were recorded.

Minimum and maximum temperatures tended to gradually increase from March to June (Figure 2). In 2021, the maximum temperature was about 14 °C degrees at the beginning of measurements in mid-March, and reached 26 °C degrees in the beginning of June, when the last measurement was recorded. However, the maximum temperature was about 15 °C degrees at the beginning of measurements in mid-March and reached 31 °C degrees in the beginning of June 2022, when the last measurement was recorded.

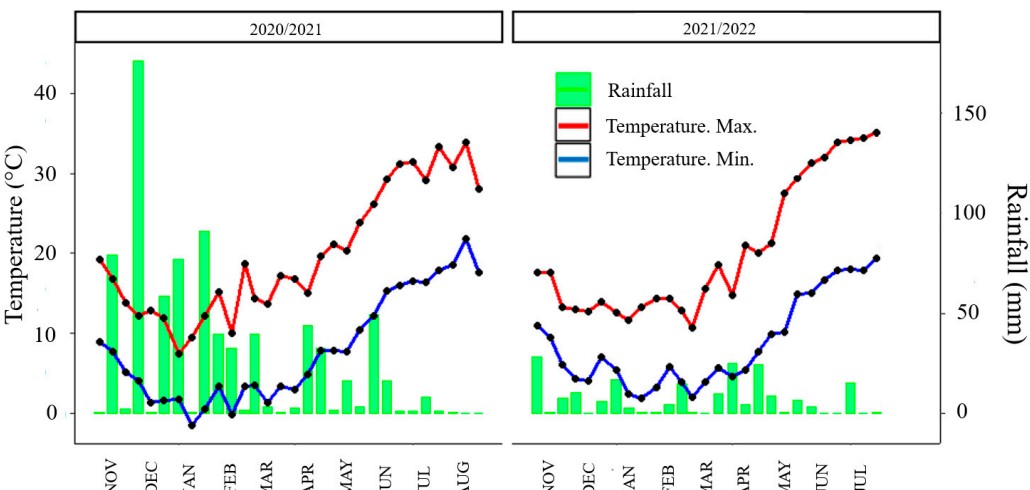

**Figure 2.** Minimum and maximum temperatures (°C), and rainfall (mm) at the experimental site, throughout the periods of study in 2021 and 2022.

### 3.2. Soil $CO_2$ Emissions, Temperature and Volumetric Water Content during the Crop Rotation
#### 3.2.1. Effect of Treatments on Durum Wheat

The results showed a similar trend for $CO_2$ emissions in both growing seasons, where it gradually increased from mid-March and reached the highest value by the end of April, and then gradually decreased until the beginning of June. In both growing seasons, soil $CO_2$ emissions differed between soil tillage treatments at the end of April, when the highest values were recorded. In 2021, soil $CO_2$ emissions observed in P and R soil tillage treatments were similar at end of April, even if the values observed were lower than emissions measured under S treatment. At the same time in 2022, R showed significantly lower emissions in comparison to those in S and P. In addition, compared to P, lower emissions were recorded in R at the beginning of June in both growing seasons.

Soil water content during the durum wheat growing seasons showed different trends under different tillage practices in both growing seasons (Figure 3).

Compared to conventional tillage, higher soil water content was observed under R treatment at all timepoints in 2020/2021. In 2021/2022 growing seasons, soil water content showed a similar trend to that observed in 2020/2021, even if the main changes were observed at the beginning of April, when the highest values were observed: R followed by S and P (Figure 3). From mid-March till the end of April, soil water content tended to be higher under both conservation tillage practices in comparison with P, even continuing to mid-May in 2021. No significant differences were detected between soil tillage systems at the beginning of June in both growing seasons.

Soil temperature tended to increase from the beginning of April until the end of the measurements in both growing seasons, even if the soil temperature growth was faster during the 2021/2022 growing season. Significant differences in soil temperature were observed from the end of April until the start of June in 2021, where higher soil temperatures were observed under both R and S tillage practices compared to P (Figure 3). No differences were observed among the soil tillage practices in the 2021/2022 growing season, except at the end of April, where P showed the highest values of soil temperature.

Mineral fertilizers tended to have lower values compared with organic fertilizer in both growing seasons, (on average 0.45 and 0.47 vs. 0.47 and 0.37 g m$^{-2}$ h$^{-1}$ in 2021 and 2022, under M and O fertilization sources, respectively). The greater differences between M and O treatments were more obvious in 2022, even if O produced significantly higher $CO_2$ emissions only at beginning of April compared to M. No significant differences were detected for soil temperature and soil water content between fertilizer sources during the study period in 2022.

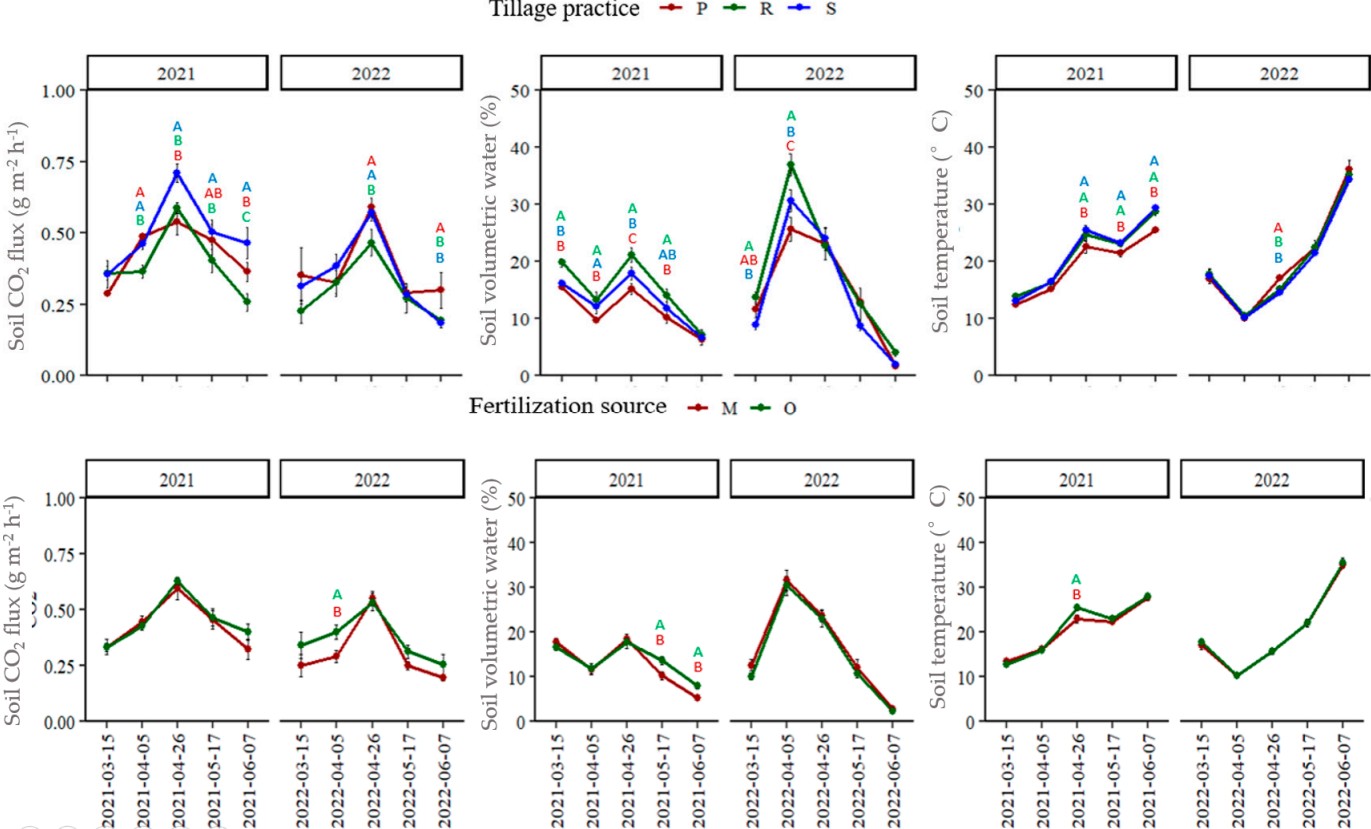

**Figure 3.** Soil CO$_2$ emissions, soil temperature, and soil volumetric water content under the three soil tillage practices: plowing (P), subsoiling (R) and spading (S), and under inorganic (M), organic (O) fertilization sources during 2021 and 2022 durum wheat growing seasons. Bars represent standard error. No letters mean that differences between the means were not significant (*p* > 0.05).

### 3.2.2. Effect of Treatments on Potato Crop

According to the results, soil CO$_2$ emissions showed various trends under the different tillage practices in both potato growing seasons. In 2021, subsoiling showed a higher peak in June than other tillage systems (on average 0.71 vs. 0.58 and 0.54 g m$^{-2}$ h$^{-1}$, respectively). Conversely, in 2022, soil CO$_2$ emissions were always lower in R compared with P and S tillage practices (on average 0.44 vs. 0.54 and 0.59 g m$^{-2}$ h$^{-1}$, respectively).

The volumetric water content of the potato was different in the two seasons, especially in the periods close to tuber sowing and harvesting. In 2021, subsoiling tended to show high values than other tillage practices, even if the greater differences were detected in July (15.66 vs. 12.25 % and 10.44 % in R, P, and S, respectively). In 2022, soil volumetric water was the highest in subsoiling compared with plowing and spading, even if the differences were observed only at the beginning of July, while in the remaining period no differences were detected among soil tillage treatments (Figure 4). In both potato growing seasons, soil temperature showed similar trends among the soil tillage treatments; it ranged from 20 °C to 30 °C during the monitoring period.

Soil CO$_2$ emission had similar trends between O and M fertilization sources, even if in both potato growing seasons the O treatments tended to show high values compared with the M treatment (Figure 4). The same period, soil water content showed a similar trend during the two growing seasons, except in 2021, when organic fertilization showed a higher value compared with mineral fertilization in the measurement period (on average 13.08 % vs. 11.24 %, respectively).

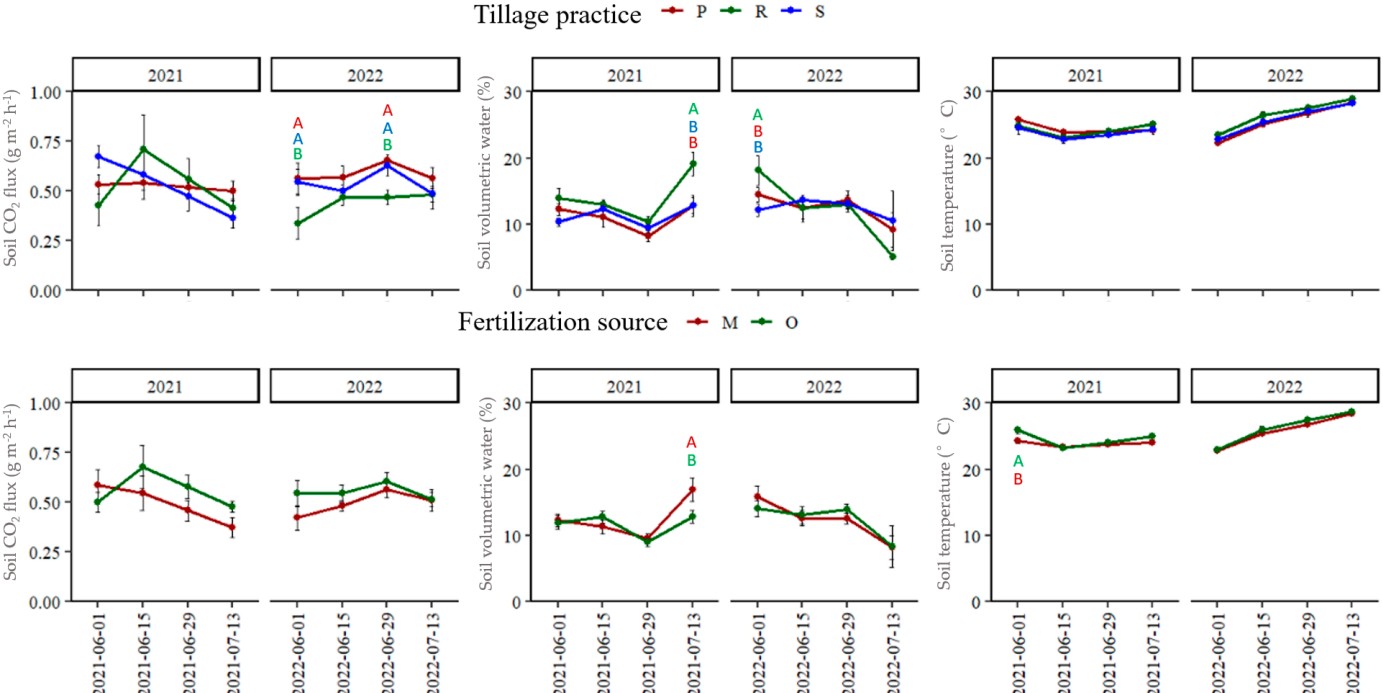

**Figure 4.** Soil $CO_2$ emissions, soil temperature, and soil volumetric water content under the three soil tillage practices: plowing (P), subsoil (R), and spading (S), and under inorganic (M), organic (O) fertilization sources during the 2021 and 2022 potato growing seasons. Bars represent standard error. No letters mean that differences between the means were not significant ($p > 0.05$).

### 3.2.3. Regression Analysis between Soil $CO_2$ Emissions with Temperature and Soil Water Content

The regression analysis between soil $CO_2$ emissions against soil temperature and soil volumetric water content were carried out for both crops, but since no differences were detected in the potato crop, only the results for durum wheat are reported (Figure 5). The results showed that the $CO_2$ emission rate was significantly dependent on volumetric water content under subsoiling and spading tillage practices only (Figure 5A). The highest level of $CO_2$ emission was 0.55 g m$^{-2}$ h$^{-1}$ (R$^2$ = 0.3296; $p < 0.001$) and was detected at 21% volumetric water content for the spading tillage system compared to subsoiling (0.48 g m$^{-2}$ h$^{-1}$; R$^2$ = 0.2883; $p < 0.001$) at 28% volumetric water content.

Regarding the fertilization source, the results also showed the highest level of $CO_2$ emissions at the optimal soil moisture level in organic fertilization (0.49 g m$^{-2}$ h$^{-1}$; R$^2$ = 0.1671; $p < 0.0001$) compared to the mineral source (0.45 g m$^{-2}$ h$^{-1}$; R$^2$ = 0.1305; $p < 0.0001$), with both at 23% volumetric water content (Figure 5B). In addition, soil temperature and water content showed significant relationships with soil $CO_2$ emissions under both fertilization sources during wheat growing seasons (Figure 5B,D). However, under equal conditions of soil water content and soil temperature, higher soil $CO_2$ emissions were obtained in organic fertilizer compared to those in inorganic fertilizer. Soil temperature and water content explained 9–16% of the variability in $CO_2$ emissions under different fertilization sources during wheat growing seasons.

The analysis of polynomial regression also showed the dependence of the $CO_2$ emission rate on soil temperature. The highest rate of $CO_2$ emission was related to spading (0.50 g m$^{-2}$ h$^{-1}$ at 21 °C soil temperature; R$^2$ = 0.1802; $p < 0.001$), the intermediate value under plowing (0.46 g m$^{-2}$ h$^{-1}$ at 23 °C, R$^2$ = 0.1039; $p < 0.022$), and the lowest value under subsoiling (0.38 g m$^{-2}$ h$^{-1}$ at 19 °C; R$^2$ = 0.1283; $p < 0.008$) (Figure 5C). Moreover, significant relationships were noticed under both fertilization sources. There was a higher value under organic fertilizer (0.47 g m$^{-2}$ h$^{-1}$; R$^2$ = 0.908; $p < 0.006$) compared to the mineral fertilizer (0.45 g m$^{-2}$ h$^{-1}$; R$^2$ = 0.1649; $p < 0.0001$, Figure 5D).

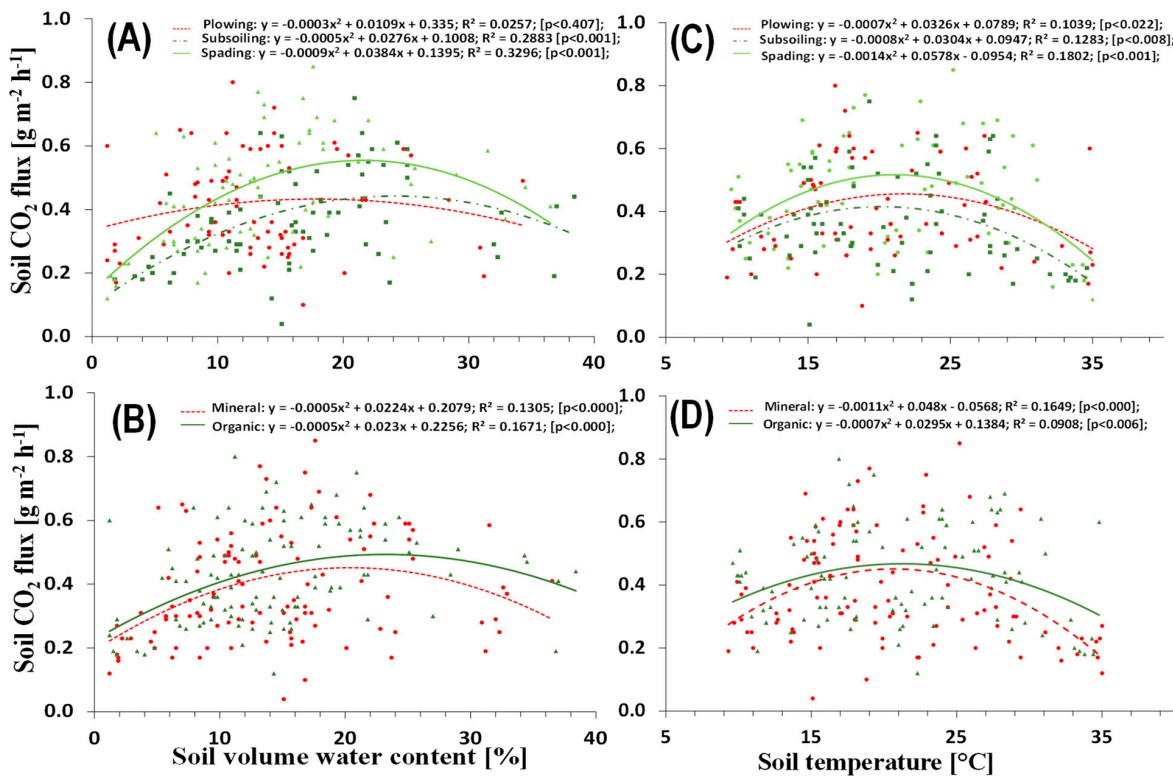

**Figure 5.** Soil $CO_2$ flux plotted against soil volumetric water content and soil temperature under soil tillage practices: plowing (P), subsoiling (R), and spading (S), and under inorganic (M), organic (O) fertilization sources for both the 2021 and 2022 durum wheat growing seasons. Soil $CO_2$ flux plotted against soil volumetric water content under different tillage (**A**) and different fertilizers (**B**). Soil $CO_2$ flux plotted against soil temperature under different tillage (**C**) and different fertilizers (**D**).

### 3.2.4. Regression Analysis between Soil $CO_2$ Emissions with Total Organic C and Total N

A linear regression was conducted for both crops, but significance was found only in the wheat crop data (Figure 6). The results of the linear regression show that there is a close relationship between the soil $CO_2$ flux emissions and the total organic carbon and the total nitrogen in the soil for wheat only. Significant relationships between soil $CO_2$ emissions and the soil nitrogen content under different soil tillage and fertilization managements were observed. However, significant relationships between soil $CO_2$ emissions and the soil carbon content (C%) were found only under fertilization treatments. Interestingly, a significant negative relationship between $CO_2$ emissions and C% was only detected under organic management ($R^2 = 0.2376$; $p < 0.039$).

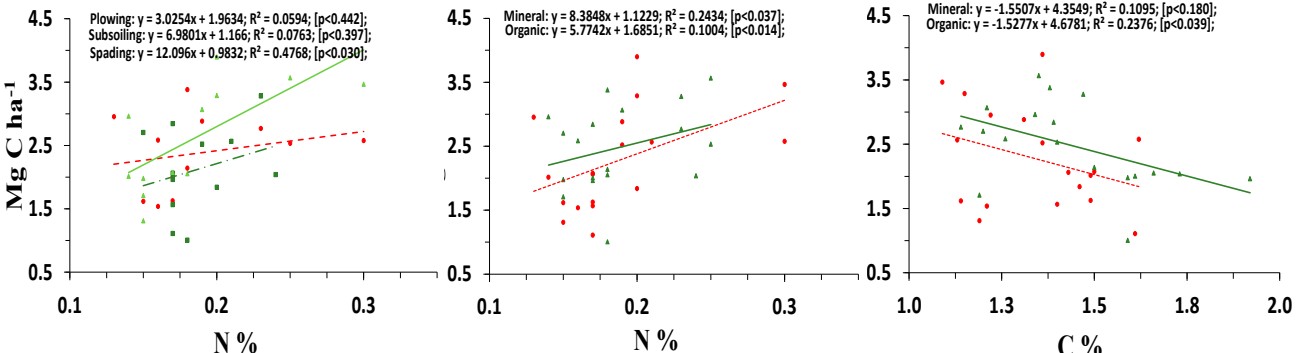

**Figure 6.** Soil $CO_2$ flux plotted in the two fertilization (Mineral - - - and Organic —) and three soil tillage (Plowing - - -, Subsoiling —, and Spading - - -) against total carbon (C%) and total nitrogen (N%) during wheat crop seasons.

However, significant positive relationships were detected between soil $CO_2$ emissions and the soil nitrogen content (N%) under both fertilization sources. This relationship was stronger under M ($R^2$ = 0.2434; $p < 0.037$) than O ($R^2$ = 0.1004; $p < 0.014$). The strongest association between soil $CO_2$ emissions and the soil nitrogen content (N%) was detected under the spading tillage system ($R^2$ = 0.4768; $p < 0.030$).

### 3.2.5. Agro-Ecosystem Carbon Balance

The C inputs were influenced by the growing seasons and fertilization source. Higher C inputs were obtained in 2021 than 2022 for both crops. Under organic management, durum wheat showed higher C inputs by biomass compared with mineral fertilizer (Table 1), while the opposite trend was observed in the potato crop (Table 2). Regarding the influence of soil tillage on C inputs, significant effects were only observed in potato cultivation, which has shown higher values in plowing (0.907 Mg C ha$^{-1}$) and spading (0.910 Mg C ha$^{-1}$) than in subsoiling (0.844 Mg C ha$^{-1}$, Table 2).

**Table 1.** The carbon balance in the wheat crop and the effects related to soil tillage practice, fertilization source and year. Different letters for each main effect indicate statistically significant differences according to the LSD test ($p < 0.05$). No letters mean that differences between the means were not significant ($p > 0.05$).

| | C Input by Biomass [Mg C ha$^{-1}$] | | C Output by CO$_2$ Emission [Mg C ha$^{-1}$] | | C Input/C Output [Mg C ha$^{-1}$] | | CO$_2$ Emission/Grain [Mg C ha$^{-1}$] | |
|---|---|---|---|---|---|---|---|---|
| **Soil Tillage** | | | | | | | | |
| Plowing | 5.139 | | 2.385 | b | 2.242 | b | 0.973 | a |
| Subsoiling | 5.660 | | 2.126 | c | 3.010 | a | 0.769 | b |
| Spading | 5.583 | | 2.617 | a | 2.331 | b | 1.043 | a |
| **Soil Fertilization** | | | | | | | | |
| Mineral | 5.236 | b | 2.274 | b | 2.545 | | 0.783 | b |
| Organic | 5.685 | a | 2.478 | a | 2.510 | | 1.074 | a |
| **Year** | | | | | | | | |
| 2021 | 5.612 | a | 2.966 | a | 1.929 | b | 1.381 | a |
| 2022 | 5.308 | b | 1.786 | b | 3.127 | a | 0.476 | b |

**Table 2.** The carbon balance in the potato crop and the effects related to soil tillage practice, fertilization source, and year. Different letters for each main effect indicate statistically significant differences according to LSD test ($p < 0.05$). No letters mean that differences between the means were not significant ($p > 0.05$).

| | C Input by Biomass [Mg C ha$^{-1}$] | | C Output by CO$_2$ Emission [Mg C ha$^{-1}$] | C Input/C Output [Mg C ha$^{-1}$] | | CO$_2$ Emission/ Dry Tuber [Mg C ha$^{-1}$] | | CO$_2$ Emission/ Fresh Tuber [Mg C ha$^{-1}$] | |
|---|---|---|---|---|---|---|---|---|---|
| **Soil Tillage** | | | | | | | | | |
| Plowing | 0.907 | a | 2.301 | 0.406 | | 4.997 | | 0.773 | |
| Subsoiling | 0.844 | b | 2.156 | 0.426 | | 5.535 | | 0.880 | |
| Spading | 0.910 | a | 2.220 | 0.419 | | 5.643 | | 0.887 | |
| **Soil Fertilization** | | | | | | | | | |
| Mineral | 0.329 | b | 2.073 | 0.171 | b | 4.597 | | 0.714 | |
| Organic | 1.445 | a | 2.378 | 0.663 | a | 6.187 | | 0.979 | |
| **Year** | | | | | | | | | |
| 2021 | 0.967 | a | 2.352 | 0.424 | | 3.000 | b | 0.550 | |
| 2022 | 0.808 | b | 2.099 | 0.410 | | 7.783 | a | 1.143 | b |

C outputs were affected by all factors for the durum wheat crop. C outputs were higher in 2021 than 2022 (2.966 vs. 1.786 Mg C ha$^{-1}$, respectively), and when using organic fertilizer compared to mineral fertilizer (2.478 vs. 2.274 Mg C ha$^{-1}$, respectively). In addition, the highest C outputs in durum wheat were observed using spading tillage followed by plowing, while C outputs were the lowest under subsoiling (Table 1). No significant differences were found on C outputs for the potato crop.

The C input/C output ratio was influenced differently between both crops. For the potato crop, the ratio was only affected by fertilization source, which was higher with an organic source (0.663 Mg C ha$^{-1}$) than mineral (0.171 Mg C ha$^{-1}$). On the other hand, for wheat cultivation, the ratio varied between growing seasons; it was higher in 2022 (3.127 Mg C ha$^{-1}$) than in 2021 (1.929 Mg C ha$^{-1}$). In addition, the ratio was affected by the soil tillage system, where subsoiling showed a significantly higher value (3.010 Mg C ha$^{-1}$) compared to other tillage practices (2.242 Mg C ha$^{-1}$ for plowing and 2.331 Mg C ha$^{-1}$ for spading, Table 1).

The ratio of $CO_2$ emissions from wheat grain yield (Table 1) had significant effects from all treatments and different growth years. The C output/grain was higher in 2021 than in 2022 (respectively, 1.381 Mg C ha$^{-1}$ and 0.476 Mg C ha$^{-1}$) and higher in organic fertilization than mineral fertilization: 1.074 Mg C ha$^{-1}$ towards 0.783 Mg C ha$^{-1}$, respectively. In addition, the subsoiling practice was distinguished from other tillage practices with a value of 0.769 Mg C ha$^{-1}$, significantly less than plowing (0.973 Mg C ha$^{-1}$) and the of the spading machine (1.043 Mg C ha$^{-1}$). However, the only factor significantly that influenced the $CO_2$ ratio emission/production in potato crop was the growing season, where 2021 showed lower values (3.000 Mg C ha$^{-1}$ and 0.550 Mg C ha$^{-1}$ for fresh and dried tubers, respectively) than 2022 (7.783 Mg C ha$^{-1}$ and 1.143 Mg C ha$^{-1}$ for fresh and dried tubers).

## 4. Discussion

The results of this work showed that agricultural practices play a significant role in dynamic of soil carbon and $CO_2$ emissions, which is consistent with previous research such as Radicetti et al. [41] and Smith et al. [42]. The effects of various tillage practices were stronger than those of fertilization sources on soil $CO_2$ emissions during wheat growing seasons. Although soil $CO_2$ emissions represent the result of very complex interactions between all biotic and abiotic factors of the agroecosystem, this finding is very important for utilizing natural resources in a sustainable way while having satisfactory agricultural production, minimal soil $CO_2$ emissions into the atmosphere, and maximal soil carbon sequestration by which climate change could be mitigated. Conservation tillage practices are well known to reduce the soil disturbance compared to that caused by plowing, promoting the right balance between micropores and macropores and slowing the rate of carbon oxidation. Therefore, any effort to decrease tillage intensity and maximize residue return should result in carbon sequestration for enhanced environmental quality [43]. However, regardless of the treatments, $CO_2$ emissions during durum wheat cultivation followed the same trend in both years, despite extremely different climate conditions. During the wheat growing season, soil $CO_2$ emissions tended to increase from mid-March till the end of April as air temperature rose, and then $CO_2$ emissions decreased until the beginning of June. A reduction in air temperature was observed in mid-April 2021 and the beginning of April 2022, followed by a raise in air temperature until the end of measurements during wheat seasons.

No significant differences were noticed in soil $CO_2$ emissions for the treatments (tillage or fertilization) in potato cropping season. Moreover, $CO_2$ emissions from organic sources tended to be higher in both growing seasons. According with the findings of Li and colleagues [44], soil $CO_2$ emissions may also result from substrate-induced priming, stimulating the decomposition of native soil C. It is well known that the availability of nutrients is essential for the vital processes of the soil. Therefore, the natural content of N and C in soil and fertilizer inputs plays a key role [18]. Organic fertilizers and compost increase SOC pools more than mineral fertilizers, so these could have a higher impact on

microbial activity [45,46]. Soil tillage practices showed seasonal differences in $CO_2$ fluxes in potato crop cycles for 2021 and 2022. In 2021, it was observed that there was a tendency under both conservative tillage systems to reduce the emissions of $CO_2$ in comparison with conventional tillage [47,48]. Stable emissions were observed throughout the potato growing seasons under plowing. However, in 2022 emissions from subsoiling were significantly lower than the conventional practice. In fact, reduced tillage practices limited the number of macro-pores, which limit the presence of air and therefore decrease $CO_2$ emissions [48]. Nevertheless, the results of this work suggest that the subsoiling tillage practice (R) could lower soil $CO_2$ emissions under both crop species. This is consistent with other work stating that minimum tillage practices reduced $CO_2$ emissions and facilitated soil carbon sequestration [49,50]. Moreover, it was reported that conservative management can be a means of conserving soil moisture and can improve water-use efficiency [51,52]. In both growing seasons of wheat, the soil water content tended to be higher under conservation tillage practices compared to the conventional system.

Significant relationships were detected between soil temperate and soil water content and soil $CO_2$ in the study. These relationships were found only for durum wheat. A significant association between soil $CO_2$ and soil water content was detected only under both conservation tillage practices during wheat growing seasons. However, the association of soil $CO_2$ with soil temperature was detected under both conventional and conservation tillage practices. Moreover, a stronger relationship was observed between $CO_2$ and soil water content than with soil temperature under different soil tillage practices. In this study, soil temperature explained 10%-18% of the variability in $CO_2$ emissions under different soil tillage practices during wheat growing seasons. Soil water content explained 29%-33% of $CO_2$ emission variability under conservation tillage practices. Despite having the same soil water content and soil temperature, higher soil $CO_2$ emissions were observed in the spading tillage system compared to subsoiling and plowing. It is often reported that soil temperature is a key regulator of soil $CO_2$ emissions [53]. However, soil temperature had no significant relationship with soil $CO_2$ emissions during potato growing seasons. This might be related to the high soil temperature (more than 20 °C) from the start of measurements in June till mid-July.

In addition, no significant difference was detected between treatments (soil tillage or fertilization managements) on soil temperature during potato growing seasons, except at the beginning of June in 2021only between fertilization sources. Although previous studies suggest as conservation tillage practices associated with soil cover contribute to reduce high soil temperatures [31,41], in this study the previous crop residues (wheat straw) are incorporated into the soil, causing a lack of insulation by residue that reduced the effects on soil temperature, determining similar values of the conventional tillage in according with the findings of Shen et al. [54]. In addition, potato hilling performed twice in all plots before the canopy closure homogenized the variation in soil temperature among treatments as observed by Nyawade et al. [55]. At the beginning of June 2021, the highest precipitation value was recorded during the time of measurements (more than 10 mm). High rainfall might be related to high soil respiration in that time of first measurements [56]. Moreover, soil water content had no significant relationship with soil $CO_2$ emissions during potato seasons. It might be related to the fact that the potato crop was under an irrigation system, which is the traditional practice in the study area.

The highest C input/C output ratio and the lowest value of the $CO_2$ emission/grain ratio were related to subsoiling tillage system compared to the other two systems in the wheat cropping system, but the input values from biomass were not significantly different from other soil tillage treatments. These data suggest that the subsoiling tillage not only produces the same crop yield as conventional practices, but it can also increase the carbon content in the soil and requires lower $CO_2$ emissions for the same production. This agrees with other studies, which state that conventionally tilled soils have higher emissions and lower amounts of carbon in soil [47,48].

The results for wheat concerning fertilization showed no significant differences between the C input/C output ratios in the two fertilizer sources, but the $CO_2$ emission/grain ratio was higher in organic fertilization. This result can be explained by the weak relationship found between $CO_2$ emissions and nitrogen in the soil, caused by a possible unbalance in the C/N ratio.

The results for potato showed significance mainly in $CO_2$ emissions due to biomass, where plowing and spading yielded statistically higher or similar results than subsoiling. Similar results were obtained by Mancinelli et al. [39]. The subsoiling could disadvantage the growth of the potato due to the greater apparent density of the soil compared to other tillage practice [57]. In the field of fertilization, the carbon input was lower in organic fertilization; In the C input/C output ratio, organic fertilization obtained higher results than mineral fertilization, but still below the value 1.

To conclude, greatest significance in potato $CO_2$ emission occurred between years. This result could be explained by the fact that the crop was grown under irrigation, which could have flattened any differences between treatments. SOC accumulation depends on several constitutional factors of the soil, including soil humidity and temperature [58]. Humidity is the most important soil parameter for gas emissions, as it directly controls the processes of microbial activity [15].

## 5. Conclusions

This study is a contribution in the field of agroecology and provides information on the relationships between traditional and conservation agronomic practices on soil $CO_2$ emissions under a durum wheat–potato rotation system, a typical cropping system in central Italy and the whole Mediterranean area.

The effects of various tillage practices were stronger than those of fertilization sources on soil $CO_2$ emissions during wheat growing seasons.

Higher soil $CO_2$ emissions were obtained in organic fertilizer compared to those in inorganic fertilizer under the study conditions.

During wheat growing seasons, subsoiling tillage not only produces the same crop yield as conventional practices, but also increases the carbon content in the soil and requires lower $CO_2$ emissions for the same production.

Significant relationships between soil temperate and soil water content and soil $CO_2$ were detected in the study. However, these relationships were found only for durum wheat. Sustainable agronomic practices (conservative tillage and organic fertilizer) included in the study showed several benefits, including increasing soil organic carbon, decreasing the demand for chemical inputs and energy, and the promotion of crop rotation. In this context, in the study's proposed rotation system, the cropping system is favored, due to its reduced soil tillage through subsoiling and organic fertilization, which can increase the soil carbon stock in the long term. The best performances in terms of agrifoods produced per $CO_2$ unit were determined by soil spading and plowing and by organic fertilization. Moreover, in the wheat crop, advantages in terms of soil carbon balance (soil C input/output) are determined in dry climate conditions, but there are also good performances in terms of agrifoods produced per $CO_2$ unit in wet climate conditions. In contrast, in the potato crop there are no advantages in terms of soil carbon balance (soil C input/output) determined by the climate, while there are good performances in terms of agrifoods produced per $CO_2$ unit in dry climate conditions.

**Supplementary Materials:** The following supporting information can be downloaded at: https://www.mdpi.com/article/10.3390/land12020326/s1, Table S1: The ANOVA for the effects of the soil tillage practice, fertilization source, and year on the C balance in the two experimental years of durum wheat; Table S2: The ANOVA for the effects of the soil tillage practice, fertilization source, and year on the C balance in the two experimental years of potato crop.

**Author Contributions:** Conceptualization, R.M., S.M., G.C., E.R.; methodology, R.M., S.M., M.A., E.R.; software, R.M., M.A. (Mohamed Allam); validation, R.M., M.A. (Mohamed Allam), E.R.; formal analysis, R.M., M.A. (Mohamed Allam); investigation, M.A. (Mohamed Allam), V.P., M.J., A.C., M.A. (Mariam Atait); data curation, R.M., V.P., M.A. (Mohamed Allam); writing—original draft preparation, R.M., M.A. (Mohamed Allam), V.P., M.A. (Mariam Atait); writing—review and editing, R.M., S.M., M.A. (Mohamed Allam), V.P., G.C., M.J., A.C., Z.A., M.M., M.A. (Mariam Atait), E.R.; visualization, R.M., S.M., M.A. (Mohamed Allam), V.P., G.C., M.J., A.C., Z.A., M.M., M.A. (Mariam Atait), E.R.; supervision, R.M., S.M., G.C., E.R. All authors have read and agreed to the published version of the manuscript.

**Funding:** This research received no external funding.

**Data Availability Statement:** Not applicable.

**Conflicts of Interest:** The authors declare no conflict of interest.

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
