# Peer review of "Durum Wheat–Potato Crop Rotation, Soil Tillage, and Fertilization Source Affect Soil CO2 Emission and C Storage in the Mediterranean Environment"

_land, doi:10.3390/land12020326_

Round 1

Reviewer 1 Report

Comments to the Authors

I have reviewed with interest your manuscript entitled „Durum wheat-potato crop rotation, soil tillage, and fertilization source affect soil CO2 emission and C storage in the Mediterranean environment submitted to the future number of Land.

This study evaluated the effects of three different tillage systems (plowing, subsoiling, and spading machine) and two types of fertilizer sources (inorganic vs. organic), on soil CO2 emissions, soil organic carbon content, soil temperature and volumetric water content. In addition, the relationships between these soil parameters in a two-year durum wheat-potato rotation system typical of the Mediterranean area will be evaluated.

In my opinion the current version of your manuscript is suitable for publication in Land, but after small revisions. The quality of the presentation should be improved. In general, manuscript is well written.

There are same grammar mistakes and awkward sentences, that have to be improved.

The article suffers from a number of small mistakes, ranging from misspellings to incorrectly phrased sentences.

Some adjustments are suggested to qualify the paper:

Issues include:

The Abstract should not exceed 200 words. Now there is 126 words. In my opinion it can be the lengthen.

General comment to the Introduction section: The content of the literature review chapter is related to the research topic. Up-to-date literature references are presented in the manuscript by the author(s).

In the chapter "Materials and Methods", the methodology is adequate.

But I have same question:

1 In the first subsection of Materials and methods titled: Study site, Authors include the data: soil texture and chemical composition but only total organic content of C and N and soil pH. Please provide other chemical parameters (P, K, Mg, humus,…), and references on the methods used to provide the soil properties.

2Sowing density of durum wheat have to be express in number grain per m2. Correct it please.

3 Why the Authors include only two years of the study. Three years of the study are more proper for statistical

analysis.

4I suggest that the characteristics of weather conditions in the analyzed vegetation periods have to be described

by the Selianinov hydrothermal index (Selyaninov's Hydrothermal Coefficient) (see: Skowera 2014, Radomski

1973, quoted in Selyaninov 1930) also called the water protection factor or conventional moisture balance. The indicator (k) determines the ratio of the sum of precipitation to the sum of average daily air temperatures in a given period:

         k =10∙P/Σt

where:

P - monthly precipitation (mm),

Ʃt - sum of average daily temperatures in a given month > 0°C.

The ranges of hydrothermal index values and its interpretation are determined depending on the k-value:

● extremely dry – k ≤ 0.4;

● very dry – 0.4 < k ≤ 0.7;

● dry – 0.7 < k ≤ 1.0;

● quite dry – 1.0 < k ≤ 1.3;

● optimal – 1.3 < k ≤ 1.6;

● moderately humid – 1.6 < k ≤ 2.0;

● humid – 2.0 < k ≤ 2.5;

● very humid – 2.5 < k ≤ 3.0;

● extremely humid – k > 3.0.

5There is also lack of information of the size of experimental plots. Could You complete it?

In the chapter "Results", the results are displayed correctly.

The “Discussion” is informative. Moreover, the Authors attempt to discuss their important results and the rest is a quotation of literature.

The Conclusions are correctly.

The list of References is enough, but a lot of Literature are not actual, before 2010 (33%). I suggest that Authors have to correct it and add same actual literature.

Same information are including in the text of Manuscript.

I hope that these comments help you to make an improved version of the manuscript.

Reviewer 2 Report

This study research the effect of different tillage methods and fertilisers on CO2 emission and C storage in Mediterranean region.

The article deals with a very interesting and important topic, but I would suggest a number of points before it can be considered for publication.

I therefore recommend a major review.

My suggestions can be found below.

Abstract

Put one more sentence more detailed result and at the end one sentence conclusion why is it important.

Introduction

Too general, more than the half of the introduction only contains general information about CO2 emission in agricultural area. I recommend shorter these parts and write a more detailed section about why these methods and why these forms of cultivation were chosen. Where and how the results can be used?

Materials and methods

In the study area section, I miss an overview map and references to the listed values (eg. geographical characterisation and soil attributes).

In the data collection section, I could recommend to use more references. I f you did not collect by your self the meteorological data, please cite.

I miss the reason why these statistical methods have been chosen for analysis, some sentence and reference needed.

Results

There are two 3.2.3 subsection.

In both 3.2.3 subsection, when you describing correlation results always sign the number of pairs also in the text and figures or tables. Until we do not see the number of pairs, we cannot say that one is stronger or weaker. Rewrite the relevant parts accordingly.

Table 1 and 2. Describe the statistical differences (a and b) in the caption and more detailed in the text.

Discussion

Between L390-405 There is only one reference, need more reference to support your results here. Last sentence also needs some reference to prove the reason why potato has no relationship with these variables.

L414-419 Just results, no conclusions. Delete this part or put some conclusion.

At the end of discussion and conclusion sections put some exact recommendation. What could the world or the region profit from these results, why is it important.

Reviewer 3 Report

Title: subscript of CO2 should be checked throughout the manuscript.

L22: Define CO2 when it appears for the first time.

L23: Give the scientific name of the plants when it appears for the first time.

Better to give information on the method in this section.

Also, better to give some numbers of highlighted results.

L60: carbon has been defined as C, please use it consistently throughout the manuscript.

L50-59: these two paragraphs could be meagered.

L61: Soil tillage practices?

L70: Better to give the definition, or at least, the form of conservative tillage techniques.

L72-77: Is there a hypothesis?

The results section has to be simplified to highlight the finding relate to the objectives or hypothesis of this study, don’t need to give all details.

L159: It's okay to descript climate data without statistical analysis, however, other data should give the significance between treatments. Check this throughout the results section.

L181: Correct the capital letters of emissions and temperature.

L185: are presented in Fig. 2, this kind of sentence should be removed to simplify the results.

L362-363: This is an important finding of this study, and should give further explanation or interpretation. For instance, further study should focus on the tillage practices to mitigate CO2 emission in this system.

L371-372, L430: Is that possible that C addition induces the SOM decomposition? Please consider this paper, 

https://doi.org/10.1111/ejss.13124

The discussion should consider citing more latest and relevant papers to back up the arguments.

L462: Limitations of this study or suggestions for further study should be given here.

Round 2

Reviewer 2 Report

The manuscript has improved a lot compared to the last version. The authors have implemented my suggestions.

Reviewer 3 Report

I appreciate the authors addressed my comments.